# Prevalence, acceptability, and cost of routine screening for pulmonary tuberculosis among pregnant women in Cotonou, Benin

**Mênonli Adjobimey**[1,2]*, **Serge Ade**[1,3], **Prudence Wachinou**[1,2], **Marius Esse**[1], **Lydia Yaha**[1], **Wilfried Bekou**[1], **Jonathon R. Campbell** [4], **Narcisse Toundoh**[1], **Omer Adjibode**[1], **Geoffroy Attikpa**[2], **Gildas Agodokpessi**[1,2], **Dissou Affolabi**[1,2], **Corinne S. Merle** [5]

**1** National Tuberculosis Program, Cotonou, Benin, **2** Faculty of Health Sciences, University of Abomey-Calavi, Cotonou, Benin, **3** Faculty of Medicine, University of Parakou, Parakou, Benin, **4** Department of Epidemiology, Biostatistics, and Occupational Health, McGill University, Montreal, Quebec, Canada, **5** Special Programme for Research & Training in Tropical Diseases (TDR), World Health Organization, Geneva, Switzerland

* menoladjobi@yahoo.fr

## Abstract

### Objectives

We sought to evaluate the yield, cost, feasibility, and acceptability of routine tuberculosis (TB) screening of pregnant women in Cotonou, Benin.

### Design

Mixed-methods, cross-sectional study with a cost assessment.

### Setting

Eight participating health facilities in Cotonou, Benin.

### Participants

Consecutive pregnant women presenting for antenatal care at any participating site who were not in labor or currently being treated for TB from April 2017 to April 2018.

### Interventions

Screening for the presence of TB symptoms by midwives and Xpert MTB/RIF for those with cough for at least two weeks. Semi-structured interviews with 14 midwives and 16 pregnant women about experiences with TB screening.

### Primary and secondary outcome measures

Proportion of pregnant women with cough of at least two weeks and/or microbiologically confirmed TB. The cost per pregnant woman screened and per TB case diagnosed in 2019 USD from the health system perspective.

**Data Availability Statement:** All relevant data are within the paper and its Supporting information files.

**Funding:** TDR, the Special Programme for Research and Training in Tropical Diseases, funded the development and conduct of this study as part of the West African Regional Network for TB control (WARN-TB) implementation research training programme. TDR can conduct its work thanks to the commitment and support from a variety of funders. These include long-term core contributors from national governments and international institutions, as well as designated funding for specific projects within our current priorities. A full list of TDR donors is available at: https://www.who.int/tdr/about/funding/en/.

**Competing interests:** CSM is currently a staff member of the World Health Organization; the author alone is responsible for the views expressed in this publication and they do not necessarily represent the decisions, policy or views of the WHO.

## Results

Out of 4,070 pregnant women enrolled in the study, 94 (2.3%) had a cough for at least two weeks at the time of screening. The average (standard deviation) age of symptomatic women was 26 ± 5 years and 5 (5.3%) had HIV. Among the 94 symptomatic women, 2 (2.3%) had microbiologically confirmed TB for a TB prevalence of 49 per 100,000 (95% CI: 6 to 177 per 100,000) among pregnant women enrolled in the study. The average cost to screen one pregnant woman for TB was $1.12 USD and the cost per TB case diagnosed was $2271 USD. Thematic analysis suggested knowledge of TB complications in pregnancy was low, but that routine TB screening was acceptable to both midwives and pregnant women.

## Conclusion

Enhanced screening for TB among pregnant women is feasible, acceptable, and inexpensive per woman screened, however in this setting has suboptimal yield even if it can contribute to enhance TB case finding.

## Introduction

Tuberculosis (TB) continues to be a major public health concern, particularly in low- and middle-income countries, despite significant efforts by the international community [1, 2]. According to the World Health Organization (WHO), there was an estimated 10 million TB cases resulting in 1.5 million deaths in 2020 [3]. Among women, TB is more common during childbearing age and is a major cause of maternal and infant mortality [4]. It is also one of the top three causes of death among women aged 15 to 49 years old [5, 6]. Pregnant women are at increased risk of TB and adverse maternal and fetal outcomes [4]. Therefore, WHO classifies pregnant women as a high risk, vulnerable population and recommends active case finding for early detection of TB [7].

The prevalence of symptoms consistent with TB and confirmed TB disease among pregnant women varies between epidemiologic contexts and with screening approaches. In the United Kingdom the incidence of TB among pregnant women was approximately 1.5-times higher than the general population [8], however in a similarly low TB prevalence setting of the United States the yield of screening was only 0.025% [9]. Yield of screening is also inconsistent in high-burden settings. In Pakistan, though 2.6% of pregnant women had symptoms consistent with TB, only 0.025% were diagnosed with TB [10]. Comparatively, in eSwatini, TB prevalence was 2% among HIV-negative pregnant women [11], while in South Africa, among pregnant women living with HIV, 16% had TB symptoms, and 2.5% had TB [12].

Symptom presentation among pregnant women also varies, which affects the sensitivity of symptom screening as a triage test for microbiologic testing. Several studies report WHO recommended symptom screening among pregnant women has sensitivity that varies from 28% to 54% [4, 12] largely driven by prolonged cough. This is lower than estimates from studies informing World Health Organization symptom screening recommendations which suggest symptom screening has a sensitivity of 73% [13, 14].

In Benin, TB mainly affects the young adult population between 25 and 44 years of age, with nearly one-third of all people diagnosed with TB being women of childbearing age [15]. However, pregnancy data is not routinely collected on TB case report forms and there are no

national guidelines for screening and management of TB in pregnant women, largely due to insufficient and inconsistent data [4]. In the absence of national guidelines and data on TB in pregnant women in Benin, this study was initiated. The objectives of the study were to implement enhanced case finding among pregnant women, to describe characteristics of women with TB symptoms, and to evaluate the yield, cost, feasibility, and acceptability of such a programme in Cotonou, Benin.

## Methods

### Study setting

This study took place in Benin, a country located in West Africa with an estimated TB incidence of 55 per 100,000 population in 2019. HIV seroprevalence in the general population was only 1.2% in 2012 [16], but the proportion of TB patients whose HIV status is known is 98% and HIV seroprevalence among TB patients has remained stable at around 16% from 2011 to 2017 [15]. As of 2018, there are approximately 370,000 pregnancies annually in Benin [17].

The city of Cotonou, with a population of approximately 685,000, was where the study was implemented. HIV prevalence in the city of Cotonou is higher than other cities at 1.9% [11] and approximately the same among pregnant women with known HIV status (1.6%) [11]. The study included eight health centers—three public, three religious, and two private facilities—collaborating with the national tuberculosis programme and conducting antenatal care (ANC) activities. All health facilities were located within a radius of 15km from the National TB hospital named Centre National Hospitalier Universitaire de Pneumo-Phtisiologie de Cotonou (CNHU-PPC)—which also houses the national tuberculosis program (NTP).

### Study design

This was a mixed-methods cross-sectional study combined with a cost assessment that took place from April 2017 to April 2018.

### Inclusion and exclusion criteria

All women presenting to participating health facilities for routine pregnancy assessments were assessed for inclusion during the study period. Inclusion criteria were: age between 14 and 45 years and evidence of pregnancy based on a positive pregnancy test, ultrasound or gynecological examination. Exclusion criteria were: women in labor or those with already diagnosed TB and/or under TB treatment at the time of the survey.

### Sample size

We calculated the sample size required to estimate the prevalence of TB using a previous study from Zambia [18] which found a TB prevalence of 1.5% among pregnant women with TB symptoms. Using a Poisson regression formula, type I error rate of 5% and type II error rate of 10%, we estimated we would need to recruit 1022 pregnant women with TB symptoms to estimate this same prevalence or a total of 4444 pregnant women regardless of TB symptoms (if based on the Zambia study, where ~23% of the pregnant women were classified as presumptive TB patients [18]).

### Study procedures

Prior to the start of the study, all midwives performing antenatal care at participating health centres received training regarding TB, its deleterious impact on pregnancy, and TB symptoms such as cough for ≥2 weeks, fever, night sweats, or weight loss. Taking into account that

prolonged cough is the primary symptom indicative of TB, NTP prioritization of persons with cough for further TB evaluation, and midwife workload, after recording all symptoms of TB, only those with cough of at least two weeks received microbiologic testing with Xpert MTB/ RIF.

Pregnant women presenting to healthcare facilities for the first time gave informed consent and were screened for all TB symptoms by the attending midwife. Symptoms were self-reported. Subsequent data on women without cough for at least two weeks were not collected. For women with a cough of at least two weeks, the midwife administered a structured questionnaire to collect data on socio-demographic, occupational, household, and clinical characteristics. All symptomatic women had sputum collected by the midwife (spot sputum). Each day a representative from the NTP would call the health facilities and come to the clinic to collect sputum samples should they have been collected that day. All sputum samples were analyzed at the CNHU-PPC using Xpert MTB/RIF. Women found to have tuberculosis were managed by standard NTP protocols. All data collection and study progress were monitored by the NTP, with monthly meetings on study status.

## Quantitative analysis

Data collected during the study were entered anonymously using Epi Data V.3.1. and analyzed with Epi Data Client v.2.0.7.22; the dataset is available in S1 Dataset. We conducted descriptive analysis to detail the characteristics of pregnant women found to have cough for at least two weeks and describe in more detail those diagnosed with TB. We estimated the TB prevalence per screening visit among pregnant women and exact binomial 95% confidence interval (95% CI).

## Qualitative analysis

We did a qualitative analysis to understand participants' experiences (both midwives and pregnant women) with TB screening during routine antenatal appointments.

**Data collection.** We recruited two pregnant women per site (16 total) and two midwives per site (14 total, as two sites only had one midwife) to conduct in-depth interviews and broaden the perspective and breadth of responses. Sample sizes were based on convenience and what was feasible. At each site, a number was assigned to each pregnant woman and to each midwife present on the day of the survey; random sampling was used to identify who would be interviewed. For both groups, the interviewer was experienced in qualitative data collection. After explaining the goal of the interview to the participant, the interviewer used a semi-structured questionnaire developed by the authors (Table 1). Interviews were conducted in French, Fon, or Yoruba and lasted 30–45 minutes for midwives and 20–30 minutes for pregnant women. The interviews were conducted onsite in a confidential location where the participant could not be overheard. There were no repeat interviews. The interviewer took notes and recorded the entire interview with the consent of the participants. The recordings were then transcribed.

**Data analysis.** The data were analyzed by topic manually (without software) and independently by a group of three people experienced in qualitative analysis who listened to the recording and read the transcript and notes beforehand. Based on their understanding, they identified themes that represented the concepts expressed in the interviews. The three-person panel reviewed the independently assessed results, and through a discussion process reached consensus on key themes.

**Table 1. Questions from in-depth interviews with midwives and pregnant women.**

| Participants | Questions |
|---|---|
| **Midwives** | 1. What does it mean to you to screen for tuberculosis in pregnant women? |
| | 2. Do you enjoy routinely screening patients for tuberculosis during antenatal visits? |
| | 3. Do you think it is possible to routinely screen pregnant women for tuberculosis as part of every midwife's routine? |
| | 4. What do you think might be the barriers to integrating routine tuberculosis screening into routine antenatal care? |
| | 5. What do you think are the practical prerequisites for integrating tuberculosis screening into routine antenatal care services? |
| | 6. The national tuberculosis programme is developing a project to integrate routine tuberculosis screening into the antenatal care visit for pregnant women. Would you support this implementation? |
| | 7. In your opinion, is routine screening of pregnant women for tuberculosis during antenatal visits feasible? |
| **Pregnant Women** | 1. What does it mean to you to screen for tuberculosis in pregnant women? |
| | 2. Does it make sense for you, your partner, and your family to screen for tuberculosis in your current pregnancy? |
| | 3. Was it a problem for you, your partner, and your family to screen for tuberculosis in your current pregnancy? |
| | 4. What do you think might be the barriers to integrating routine tuberculosis screening into antenatal care for pregnant women? |
| | 5. In your opinion, is routine screening of pregnant women for tuberculosis during antenatal visits acceptable? |

## Cost analysis

We did a cost analysis to estimate the additional cost of the TB screening program among pregnant women during antenatal care. We employed a microcosting approach, which considered all costs associated with the TB screening program that were in excess of the current standard of care (no TB screening), excluding costs associated with research. All costs were from the health system perspective and expressed in 2019 USD. Costs were locally collected where possible. When necessary, we converted salaries using purchasing power parity and material costs using exchange rates [19, 20].

We considered the following costs: the costs associated with training midwives in TB symptom screening and sputum collection; the costs associated with contacting facilities about the need for sputum sample pickup; the costs associated with TB symptom screening; the costs associated with sample transport; and the costs associated with Xpert MTB/RIF analysis.

To estimate personnel costs, we collected average annual salaries of midwives and laboratory technicians. We used time and motion techniques to estimate the time required to perform symptom screening (in absence and presence of symptoms) and the time required to analyze samples with Xpert MTB/RIF. For midwives, a midwife from each site was randomly selected and monitored during their workday for one week to record different activities performed and the time required for each activity. For laboratory technicians, a technician was observed receiving, preparing, and analyzing a set of sputum samples with Xpert MTB/RIF. To estimate material costs, we used the Global Drug Facility for the costs of Xpert MTB/RIF cartridges and machines. The capital costs of the machine were annualized over a five-year useful lifetime at 3% per annum and averaged on a per sample basis. Costs associated with Xpert MTB/RIF maintenance came from laboratory expense logs and were averaged on a per sample basis. Laboratory overhead costs were estimated based on annual expense logs and averaged based on manpower allocation to Xpert MTB/RIF and on a per sample basis. Since TB

screening occurred during routine antenatal care visits, we did not consider overhead costs of these visits. For costs of training and communication, we used the incurred costs of these components during the research study.

We calculated the overall cost of our TB screening intervention during the study period, and estimated: the cost per pregnant woman screened, the cost per pregnant woman identified with cough of at least two weeks, and the cost per pregnant woman diagnosed with TB.

### Ethical considerations

The study was jointly authorized by Directorate of Maternal and Child Health and the NTP, two departments of the Ministry of Health of Benin, and by the managers of the health facilities. The approval of the National Committee of Ethics for Health Research (CNERS) was obtained under the number n˚ 029 of 09/09/2016.

Written consent was obtained from all pregnant women and midwives. The consent forms were approved by the ethics committee. Information relating to the state of health of pregnant women has been treated with respect for confidentiality and the human person.

## Results

Between April 2017 and April 2018, a total of 4,070 consenting pregnant women underwent TB screening during routine antenatal care visits at eight health facilities in Cotonou, Benin (Fig 1).

### Quantitative analysis

**Characteristics of women screened and with cough of at least two weeks at their screening visit.** The 4,070 pregnant women participating in the study had an average (standard deviation) age of 27 ± 5 years. Of these, 94 women (2.3%) had cough of at least two weeks and were further assessed. They ranged in age from 16 to 40 years with an average of 26 ± 5 years (Table 2). The vast majority (77/94; 82%) had received BCG vaccination. On average, women were on their third pregnancy—75 (79%) had at least two previous pregnancies and 36 (38%) had given birth to at least two children. Among women with cough of at least two weeks, the majority (54/94; 57.4%) worked in sales and services occupations, 4 (4.2%) were exposed to second-hand smoke, and 5 (5.3%) had known previous tuberculosis contact (Table 3). The co-morbidities found among pregnant women were anemia (11/94; 11.7%), HIV (5/94; 5.3%); hepatitis B (3/94; 3.2%), and hypertension (2/94; 2.1%). Among women with a cough of at least two weeks, most had productive (i.e., with expectoration) coughs (67/94; 71.3%). The next most common symptom was fever (28/94; 29.8%), followed by weight loss/inability to gain weight (12/94; 12.8%), night sweats (9/94; 9.6%), and hemoptysis (5/94; 5.3%).

**Prevalence of TB among pregnant women with cough of at least two weeks at their screening visit.** All 94 women with a cough provided sputum for Xpert MTB/RIF analysis. Of these, 2 (2.3%) were found to have bacteriologically confirmed TB. This is equivalent to a prevalence of 49 cases per 100,000 (95% CI: 6 to 177 per 100,000) pregnant women screened per screening visit (2 cases among 4070 pregnant women screened). Both women were HIV-negative and initiated TB treatment—one in the first trimester and one later in pregnancy after being initially lost to follow-up—and delivered their children. However, the child of the mother who initiated treatment later in pregnancy died two weeks after birth; the exact cause of death is unknown to the study team.

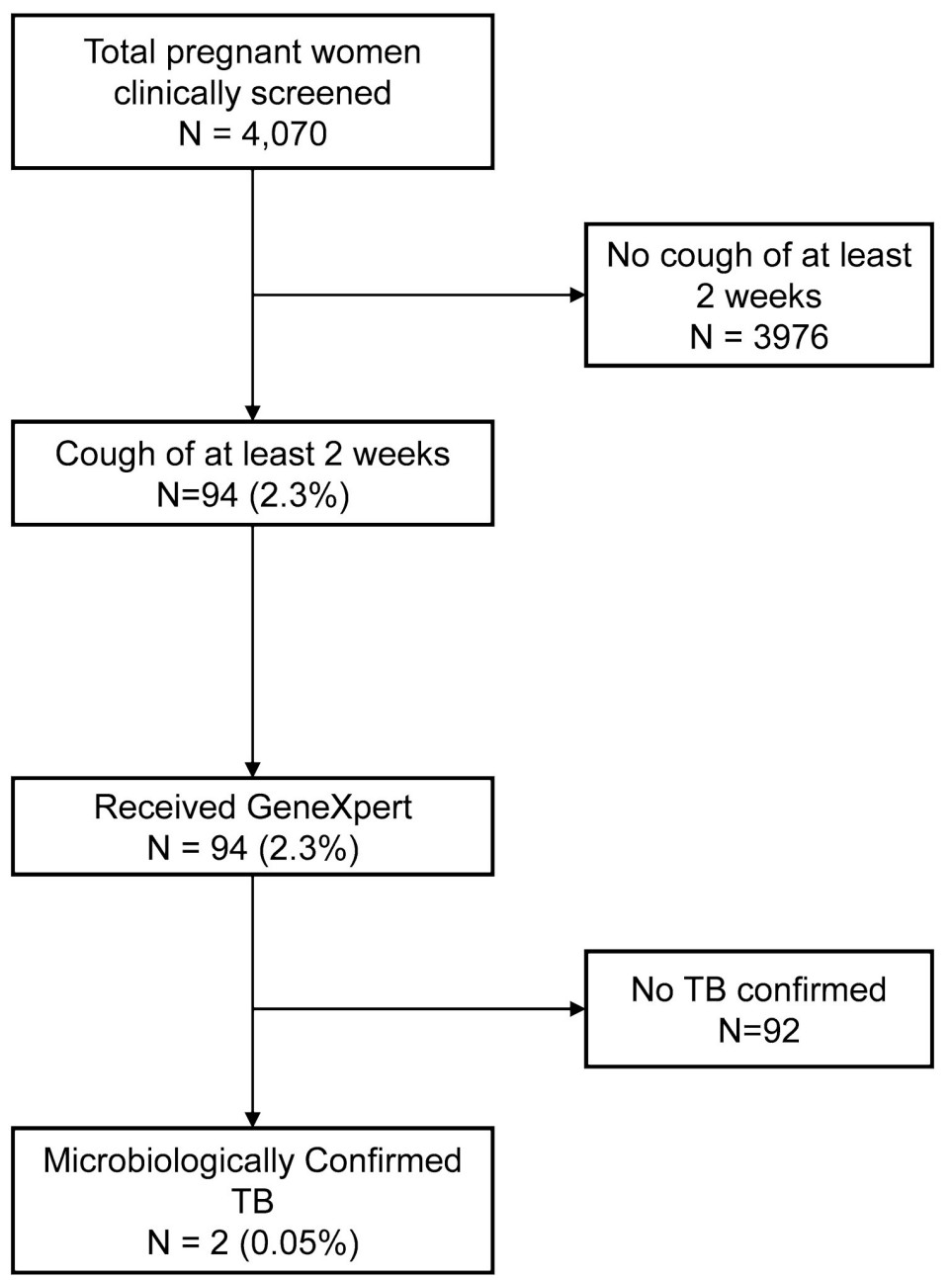

**Fig 1. Flow diagram of participants through the study.**

## Qualitative analysis

We interviewed a total of 16 pregnant women and 14 midwives for our qualitative assessment. We found that most participants had good knowledge of TB symptoms such as cough, fever, and hemoptysis.

Original Participant Response: *"Celui qui tousse et crache du sang doit courir pour aller à l'hopital, la tuberculose est souvent là"*

English Translation: *"The one who coughs, and spits blood must run to the hospital, tuberculosis is often there"*

**Table 2. Characteristics of pregnant women found to have cough of at least two weeks during antenatal care visits (N = 94).**

| | | Number | Percentage (%) |
|---|---|---|---|
| **Age** | 15–24 | 33 | 35.1 |
| | 25–34 | 52 | 55.3 |
| | 35–44 | 9 | 9.6 |
| **Level of education** | None | 27 | 28.7 |
| | Primary | 27 | 28.7 |
| | Secondary | 33 | 35.1 |
| | Post-Secondary | 7 | 7.4 |
| **Marital status** | Free Union | 31 | 33.0 |
| | Married | 59 | 62.8 |
| | Single | 4 | 4.3 |
| **Number of pregnancies** | First pregnancy | 25 | 26.6 |
| | 2–3 pregnancies | 41 | 43.6 |
| | $\geq$ 4 pregnancies | 28 | 29.8 |
| **Number of deliveries** | Zero deliveries | 19 | 20.2 |
| | One delivery | 39 | 41.5 |
| | 2–3 deliveries | 30 | 31.9 |
| | $\geq$ 4 deliveries | 6 | 6.4 |
| **Age of pregnancy** | First trimester | 36 | 38.3 |
| | Second trimester | 39 | 41.5 |
| | Third trimester | 19 | 20.2 |
| **Total** | | **94** | **100** |

Original Participant Response:*"On reconnait une personne qui a la tuberculose lorsque tousses, crache a le corps et devient blanc"*

English Translation: *"You can recognize a person who has tuberculosis when they cough, spit on their body and turns white"*

Midwives were aware of the possibility of TB complications in pregnant women, but not specific manifestations.

**Table 3. Profession, smoking status, medical history, and tuberculosis symptoms present among pregnant women with cough of at least two weeks during antenatal care visits (N = 94).**

| | | Number | Percentage (%) |
|---|---|---|---|
| **Professional and environmental features** | Sales/Service Profession | 54 | 57.4 |
| | Second Hand Smoking | 4 | 4.2 |
| | Known Previous Tuberculosis Contact | 5 | 5.3 |
| **Medical history** | Presence of BCG scar | 77 | 81.9 |
| | Anemia | 11 | 11.7 |
| | HIV | 5 | 5.3 |
| | Hepatitis B | 3 | 3.2 |
| | Hypertension | 2 | 2.1 |
| | Previous Tuberculosis Diagnosis | 1 | 1.1 |
| **TB symptoms** | Productive cough $\geq$ 2 weeks | 67 | 71.3 |
| | Fever | 28 | 29.8 |
| | Weight stagnation / weight loss | 12 | 12.8 |
| | Night sweats | 9 | 9.6 |
| | Hemoptysis | 5 | 5.3 |

Original Midwife Response:"*A force de tousser la femme est fatiguée et son bébé peut tomber malade*"

English Translation: "*Coughing makes the woman tired and her baby may get sick*"

Original Midwife Response: "*Je ne connais pas les spécificités de la tuberculose chez une femme enceinte mais la tuberculose doit être plus grave avec une grossesse*"

English Translation: "*I don't know the specifics of tuberculosis in a pregnant woman, but tuberculosis must be more serious with a pregnancy*"

The airborne mode of transmission was known by all respondents, transplacental transmission was known only by midwives. Transmission by digestive tract was less mentioned by both midwives and pregnant women.

Original Participant Response: "*La tuberculose s'attrape dans l'air qui est respiré*"

English Translation: "*TB is caught in the air you breathe*"

Original Midwife Response: "*Une maman peut donner la tuberculose à son bébé dans le ventre*"

English Translation: "*A mother can give tuberculosis to her baby in the womb*"

All pregnant women interviewed supported TB screening during routine antenatal care visits. All midwives also found routine TB screening during antenatal care visits like an opportunity to detect more TB cases, but the additional workload was also mentioned as a potential barrier and therefore a factor to be taken into consideration if TB screening was included in ANC routine care.

Original Participant Response: "*Si on peut vite dépister la tuberculose chez la maman son bébé sera plus en sécurité*"

English Translation: "*If we can quickly detect tuberculosis in the mother, her baby will be safer*"

Original Midwife Response: "*La recherche de la tuberculose pendant la grossesse est une bonne initiative mais faut pas que cela soit encore une charge additionnelle pour nous les sages-femmes, c'est pourquoi il faudra penser l'intégrer directement sur la carte maternelle de suivi des CPN*"

English Translation: "*The research of tuberculosis during pregnancy is a good initiative but it should not be an additional burden for us midwives, that's why we should think about integrating it directly on the maternal card of ANC monitoring*"

## Cost analysis

Table 4 reports the individual component costs of the TB screening intervention. Across all 4,070 pregnant women screened, the overall cost of the intervention was $4542 USD. Microbiological analysis with Xpert MTB/RIF accounted for nearly 40% of all costs and training of midwives accounted for nearly 30%. Overall, we calculated a cost of $1.12 USD per pregnant woman screened, $48.32 USD per pregnant woman identified with cough of at least 2 weeks, and $2271 USD per case of TB diagnosed.

## Discussion

In this study of 4,070 pregnant women, the prevalence of cough of at least 2 weeks was 2.3%. Among the 94 pregnant women with cough, two had bacteriologically confirmed TB for an overall prevalence of 49 per 100,000 per screening visit. The cost per pregnant woman screened was inexpensive and both pregnant women and their midwives found screening to be acceptable.

In general, symptom screening is less sensitive in pregnancy and this can lead to an underestimation of presumptive TB cases [4, 12]. In our study, only women with cough of at least 2

**Table 4. Cost components for the cost assessment.**

| Component Costs | Unit Cost (2019 USD) | Description |
|---|---|---|
| **Administrative Costs** | | |
| **Training** | **$1315** | Cost used to train midwives at each site |
| **Telephone monitoring** | **$336** | Phone fees paid $28 USD per month over the 12-month study for communication with each site |
| **Screening Costs** | | |
| **Verbal Screening for Cough** | **$0.12** | Midwife salary ($0.12 USD/min) multiplied by the time to ask about symptoms (1 minute per person) |
| **Additional Screening and Sputum Collection from Pregnant Women with Cough of at least 2 weeks** | **$1.80** | Midwife salary ($0.12 USD/min) multiplied by the time for additional screening, sputum collection, and sample registering (15 minutes per person). |
| **Transport of sputum to lab (per sample)** | **$4.70** | Transportation costs per roundtrip pickup of samples from sites. |
| **Xpert MTB/RIF (per sample)** | **$19.06** | Sum of components below |
| Cartridge and shipping costs (per sample) | $11.26 | Global Drug Facility cost of Xpert MTB/RIF cartridge |
| Technician time (per sample) | $0.59 | Technician Salary ($0.09 USD/min) multiplied by time to accession, prepare, analyze, and report each sample (6.5 min per sample, when 16 prepared at once) |
| Capital Cost of Equipment (per sample) | $1.75 | Annuitized cost of equipment (at a rate of 3%) for an expected useful life of 5 years |
| Equipment Maintenance Costs (per sample) | $0.10 | Annual maintenance costs based on national tuberculosis programme contract |
| Laboratory Overhead Costs (per sample) | $5.36** | Laboratory overhead costs estimated based on laboratory manpower and overhead expenditures from the hospital |

¶The CNHU-PPC has 2 Xpert MTB/RIF machines of 16 cartridges (144,000 USD). The lifetime of one machine is 5 years and performs 17,923 analyses per year. The $1.75 USD represents the cost adjusted over the lifetime of the machine per sample, annuitized at a 3% rate.

**The cost of overhead associated with Xpert MTB/RIF, based on the overhead of the hospital and an overhead distribution derived from the number of people employed in the laboratory dedicated to Xpert MTB/RIF compared to the rest of the hospital (6/113).

weeks gave sputum for Xpert MTB/RIF. The rate of cough found in our study was lower than that reported in Zambia which found 23% of pregnant women with cough [18], as well as in Kenya which found a prevalence of any symptoms of TB to be 8% among pregnant women living with HIV and 5% among HIV-negative pregnant women [14]. Differences may be explained by epidemiologic contexts and different frequencies of respiratory illness and therefore cough in these populations. However, our findings are similar to those found in Burkina-Faso (3%)—a country with a similar prevalence of HIV to Benin—as well as Pakistan; both studies were performed in the context of systematic screening for tuberculosis during antenatal consultation [10, 21].

The prevalence of pulmonary TB per visit among pregnant women was 49 per 100,000. Comparatively, TB incidence among a similarly aged female population in Benin from 2016–2018 was 28 per 100,000 [15, 22]. Other analyses of TB prevalence have found prevalence may be higher among pregnant women compared to the general population [6]. At the TB rates estimated in this study and an estimated 370,000 pregnancies occurring annually in Benin

[17], a screening program for all pregnant women would cost $414,000 USD and detect about 182 cases of TB.

The diagnosis of TB in both participants was made in the first trimester. Indeed, women in early pregnancy are twice as likely to develop TB as non-pregnant women [23], however risk remains throughout pregnancy and even in the postpartum period [4, 24]. Clinically, both patients had similar symptoms, which were like those found in HIV-negative pregnant women in another study [14]. In terms of treatment, only one of the two participants started anti-tuberculosis drugs early, with good adherence. Sobhy et al [25] have shown that outcomes among pregnant women tend to improve the earlier treatment starts. In their study, women treated early in the first trimester of pregnancy had no premature births, low birth weight babies, or perinatal deaths, whereas in women treated in the second or third trimester of pregnancy, 33% of infants were premature, 61% of infants had low birth weight, and 23% had perinatal deaths [25]. Indeed, the participant in our study who started treatment later in pregnancy experienced perinatal death, though it is uncertain whether this death was a result of late initiation of effective TB therapy.

Midwives' and pregnant women's knowledge of TB was good for routine TB symptoms but low about TB complications in pregnancy. This suggests awareness and information sessions may be required to improve knowledge on TB complications. The feasibility and acceptability of implementing TB screening was considered good among both pregnant women and their midwives, though it was stressed it should not be an additional burden. In 2018, the NTP of Benin offered free access to Xpert MTB/RIF as a first-line diagnostic for key populations, including pregnant women, which supports the feasibility and acceptability of such a screening programme.

We found the cost of screening pregnant women for TB was low, however within our population the yield was suboptimal. Early screening and detection of TB with prompt treatment initiation, helps minimize the adverse consequences of TB in the mother and infant [4, 5] and prevent transmission. The benefits may extend to the midwives and the health system because it may reduce strain associated with caring for pregnant women in emergency situations. The cost per TB case detected was high compared to other case-finding interventions in Benin. For example, an intervention implemented through TB REACH, in which the overall objective was to increase the detection rate of TB in populations with limited access to TB services through outreach clinics, estimated a cost of ~$610 USD per TB case diagnosed [26]. However, to formally compare different interventions, a cost-utility analysis would need to be performed to determine if any intervention is cost-effective and which should be prioritized. For pregnant women attending ANC appointments, questioning about cough can be done quickly and easily. In other populations at increased risk of TB—such as people living with HIV—taking advantage of routine care appointments to screen for TB is already done. Similarly integrating such screening for pregnant women could avert morbidity and mortality, with minimal additional effort.

## Strengths and limitations

The main strengths of this study were the diverse nature of women attending antenatal care and pragmatic implementation of the intervention reflecting integration into routine care. However, this study also has limitations. The method to identify women for further microbiologic evaluation was only cough of at least two weeks [27]. This is a less sensitive approach to TB detection than considering all possible TB symptoms for further evaluation—which already has lower sensitivity in pregnant women—however, was implemented because this is the policy of the NTP. We speculate our overall yield may have increased had we performed Xpert

MTB/RIF on all women with any TB symptom, but would result in an increase in the cost per pregnant woman screened. We found less symptomatic pregnant women than initially expected, which underpowered our estimates. This could be due to the lower prevalence of both cough and TB in Benin compared to other settings where this screening strategy has been evaluated. The absence of radiography in HIV-positive women is also a limitation. Indeed, on the basis of the symptomatology of productive cough, very few HIV-positive women were further evaluated for TB. Our study was cross-sectional in nature and only accounted for one screening event per pregnant woman. There is a possibility that women may have developed symptoms later and had TB or that women presented with subclinical TB that would be missed by TB symptom screening. We only used Xpert MTB/RIF to diagnose TB. While this is a rapid, molecular-based diagnostic test, it is less sensitive than culture—particularly among smear-negative cases—so some cases of TB may have been missed [28].

## Conclusion

In this study, we found about 1 in 40 pregnant women had prolonged cough and the prevalence of TB per screening visit was 49 per 100,000. TB screening in ANC seems to be acceptable to both midwives and pregnant women, with a low cost per woman screened. In settings where access to care is limited, ANC visits may be a useful opportunity to perform TB screening. Although the yield of the intervention was low, it must be weighed against the increased risks of adverse maternal and fetal outcomes if a pregnant woman develops TB and against other case finding interventions.

## Supporting information

**S1 Dataset.**
(XLSX)

## Acknowledgments

Our thanks go to the midwives, the pregnant women, participating sites, and the NTP of Benin.

**Disclaimer**: CSM is currently a staff member of the World Health Organization; the author alone is responsible for the views expressed in this publication and they do not necessarily represent the decisions, policy or views of the WHO.

## Author Contributions

**Conceptualization:** Mênonli Adjobimey, Serge Ade, Prudence Wachinou, Wilfried Bekou, Narcisse Toundoh, Omer Adjibode, Geoffroy Attikpa, Gildas Agodokpessi, Dissou Affolabi, Corinne S. Merle.

**Data curation:** Mênonli Adjobimey, Marius Esse, Lydia Yaha, Jonathon R. Campbell, Narcisse Toundoh.

**Formal analysis:** Mênonli Adjobimey, Marius Esse, Wilfried Bekou, Jonathon R. Campbell.

**Funding acquisition:** Corinne S. Merle.

**Investigation:** Mênonli Adjobimey.

**Methodology:** Mênonli Adjobimey.

**Project administration:** Mênonli Adjobimey.

**Resources:** Mênonli Adjobimey.

**Supervision:** Mênonli Adjobimey, Corinne S. Merle.

**Writing – original draft:** Mênonli Adjobimey, Jonathon R. Campbell.

**Writing – review & editing:** Mênonli Adjobimey, Serge Ade, Prudence Wachinou, Marius Esse, Lydia Yaha, Wilfried Bekou, Jonathon R. Campbell, Narcisse Toundoh, Omer Adjibode, Geoffroy Attikpa, Gildas Agodokpessi, Dissou Affolabi, Corinne S. Merle.

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
