## [Decision Letter · Decision Letter 0]

9 Sep 2021

PONE-D-21-23077Prevalence, acceptability and cost of routine screening for pulmonary tuberculosis among pregnant women in Cotonou, BeninPLOS ONE

Dear Dr. Campbell,

Thank you for submitting your manuscript to PLOS ONE. After careful consideration, we feel that it has merit but does not fully meet PLOS ONE’s publication criteria as it currently stands. Therefore, we invite you to submit a revised version of the manuscript that addresses the points raised during the review process.

The authors present a mixed-methods cross-sectional study assessing the implementation and costs of integrated screening for symptomatic TB disease at routine antenatal visits at 8 facilities in Benin. Pregnant and post-partum women are at high risk of TB disease and the screening initiative and the report are of potential clinical and programme relevance. The manuscript is clear and well-written. However, there are some elements of the methodology that require clarification if the data presented are seen to support the interpretation and conclusions that have been presented.

Reviewers with both quantitative and qualitative expertise have contributed comments. Please see my further remarks below, *Additional Editor Comments.*

We look forward to receiving your revised manuscript.

Kind regards,

Emma K. Kalk

Academic Editor

PLOS ONE

Journal Requirements:

https://journals.plos.org/plosone/s/file?id=ba62/PLOSOne_formatting_sample_title_authors_affiliations.pdf\\

2. Please provide additional details regarding participant consent. In the ethics statement in the Methods and online submission information, please ensure that you have specified what type you obtained (for instance, written or verbal, and if verbal, how it was documented and witnessed). If your study included minors, state whether you obtained consent from parents or guardians

Additional Editor Comments (if provided):

1. Context

1a. There are limited data on TB screening in pregnancy but the authors omit some previous studies and do not provide detail with respect to WHO recommendations:

Schwartz, et al. Pasipamire et al. (reviewer 2), Ranaivomanana et al. (Reviewer 3) and Hoffman et al. (doi:10.1371/journal.pone.0062211)

The latter found a very low sensitivity for the WHO 4-symptom screen in pregnancy which should be incorporated into your discussion/limitations. In addition, GeneXpert screening at first antenatal visit has recently been introduced in South Africa (the symptom screen has been standard of care for some time) and there may be additional data available from this country. GeneXpert may be less sensitive when used as a screening test as opposed to a diagnostic test. However, here it is used for confirmation. Prof Neil Martinson has done a lot of work in this regard.

1b. The information about TB and HIV in Benin and Conotou is helpful to situate the study; antenatal HIV prevalence rates would be relevant. Most TB in sub-Saharan Africa presents in people living with HIV. HIV status should also be presented in the results and noted in the discussion.

2. Interpretation

The antenatal TB prevalence was relatively low, although higher than that in the general population. Please note the concerns of reviewers 1,2 and 3 with respect to screening and timepoints. Some of these comments can be addressed as limitations. Most maternal TB presents in the post-partum period and screening in this population may be useful. There is also South African data on TB prevalence in pregnancy that may be useful.

It is not clear that the infant death was related to maternal TB. Be careful of making this assumption without evidence.

3. Qualitative Component

Did you use a software package to assist with analysis?

Reporting of the qualitative results is inadequate (reviewer 1).

Reviewers' comments:

Reviewer's Responses to Questions

**Comments to the Author**

1. Is the manuscript technically sound, and do the data support the conclusions?

Reviewer #1: Partly

Reviewer #2: Partly

Reviewer #3: Yes

2. Has the statistical analysis been performed appropriately and rigorously? 

Reviewer #1: No

Reviewer #2: No

Reviewer #3: I Don't Know

3. Have the authors made all data underlying the findings in their manuscript fully available?

Reviewer #1: No

Reviewer #2: No

Reviewer #3: Yes

4. Is the manuscript presented in an intelligible fashion and written in standard English?

Reviewer #1: Yes

Reviewer #2: Yes

Reviewer #3: Yes

5. Review Comments to the Author

Reviewer #1: Manuscript Number PONE-D-21-23077

Prevalence, acceptability and cost of routine screening for pulmonary tuberculosis

among pregnant women in Cotonou, Benin

Thank you for asking me to review the manuscript ‘Prevalence, acceptability and cost of routine screening for pulmonary tuberculosis among pregnant women in Cotonou, Benin.’ In this mixed-methods, cross-sectional study with a cost assessment, the authors evaluate the yield, cost, feasibility, and acceptability of routine tuberculosis (TB) screening of pregnant women in Cotonou, Benin. The manuscript is well-written and on the whole reads easily. However, I have some comments below which do need to be addressed before this manuscript can be published.

Main Comments

1. Prevalence rate

1.1. The authors’ report a TB prevalence rate of 49 per 100,000 among pregnant women enrolled in their study. However, in this cross-sectional study ~ 4000 pregnant women were screened only once during their pregnancy for TB, and not throughout their pregnancy. In addition, only those who had a cough longer than two weeks were asked to provide sputum for analysis. So, in fact, are the authors not are reporting a prevalence rate of 49 per 100,000 for one visit amongst those who had a cough and for whom a sputum sample was sent? For a TB prevalence rate among pregnant women, all ~ 4000 pregnant women would need to be screened and tested at each and every ANC visit throughout their pregnancy. The authors do acknowledge in the limitations that participants were only screened once during their pregnancy, but the authors need to clarify that they are reporting a prevalence rate of 49 per 100,000 for one visit amongst those who had a cough and for whom a sputum sample was sent.

1.2. Given the limitations in using only two cases of TB to determine a prevalence rate, a more robust discussion of the sensitivity and specificity of a cough of longer than two weeks and GeneXpert would be helpful. Given the increasing awareness of asymptomatic TB, this must be addressed in the discussion.

2. The death of the infant in one of the women diagnosed with TB.

In the all-important first paragraph of the discussion the authors write ‘the benefit of early detection and prompt treatment is apparent, given the adverse fetal outcome among one woman diagnosed with TB.’ The implication is that the infant’s death was TB-related. However, no evidence is given confirming this. Was the infant diagnosed with congenital TB? Were congenital or other morbidities excluded?

3. Reporting of cost analysis

Once published, this article will most probably be read by people who work in TB, many of whom may not be familiar with the technical terms used to describe the cost analysis. Simpler more accessible terminology should be used for the following:

‘purchasing power parity’

‘material costs’

‘accessioning’

‘annuitized’ – the term annualised is more intuitive.

‘prorated’

‘realized costs’

4. The reporting of the qualitative results (Page 10)

‘All pregnant women interviewed supported TB screening during routine antenatal care visits and

mentioned its usefulness in preventing pregnancy complications. All midwives also found routine

TB screening during antenatal care visits to be very useful…’

This very generalised reporting of the qualitative results is inadequate and more detail is needed to substantiate the findings reported.

• What do you mean pregnant women supported TB screening? What was said in the interviews that led you to this conclusion?

• What pregnancy complications did pregnant women say could be avoided by screening?

• What did the midwives say that led the authors to conclude that screening during ANC visits was useful?

Minor Comments

Page 3:

The sentence ‘Additional factors such as human immunodeficiency virus (HIV) and diagnosis of advanced disease at late term in pregnancy are also associated with poor prognosis in pregnant women with TB.’

This sentence is unclear, are you referring to advanced TB or advanced other diseases?

Page 4:

‘…..located within a radius of 15km from the NTP and included three public…..’

Do you mean the NTP offices?

‘Consecutive women…’ Do you mean all women?

‘Included women were between the ages of 14 and 45 years and had a pregnancy test, an ltrasound, or a gynecological examination certifying pregnancy…..’

Were these your inclusion and exclusion criteria? If so clarify this.

Page 5:

‘Taking into consideration midwife workload, we asked them to at least focus on the most important sign which is the presence of a cough for more than 2 weeks.’

After reading this sentence I was surprised to read in the results section that you did ask and record other symptoms. Maybe you could clarify in the text exactly what was done, something like: 'Having recorded all TB symptoms, the questionnaire focussed on coughs of more than 2 weeks.’

Page 6:

‘All participants were randomly selected.’

How were participants randomly selected?

‘The number of participants was determined a priori.’

On what basis was this a priori decision made.

‘These interviews were conducted onsite but separate from antenatal care rooms.’

What is of importance and interest is whether the interviews were conducted in a confidential place where the participant could not be overheard.

Page 9

‘Weight loss/stagnation’

By stagnation do you mean a failure to gain weight?

Page 12

‘….33% were premature…’

This is unclear, you need to clarify that this was 33% of the infants.

Reviewer #2: Introduction:

- Introduction does not mention previous studies that looked at TB screening in pregnancy e.g. Schwartz American Journal of perinatology 2009, Kosgei Public Health Action 2013, Feroz Ali International Journal of Infectious Diseases, Gounder JAIDS 2012, Pasipamire African Journal of Lab Medicine 2020 (These all include HIV-neg participants as well). I think its very important to look at what others have tried before even if it is in different settings - these differences can then be highlighted. You mention the WHO but dont telll us exactly what they recommend.

- You should also provide some information about TB in Benin - Whats the prevalence in the general population and in pregnancy? If its not known then that should be said. What are the current procedures?

- This info could lead to why you are doing this study now? At the moment its hard to understand why this study is being done now because its not clear what we know and where the gaps in knowledge are.

Methods:

- Under study population, you mention ultrasound, gynae exam etc. Were these study procedures or part of routine care? I'm not sure if I missed it but are they relevant to the study - dont think you mention any of these results.

- I think the decision to ask only about cough of 2 weeks is really important. The could affect the sensitivity and specificity. As it is, I think the 4 symptom screen might miss alot of cases especially in pregnancy when TB symptoms can be masked/atypical. The sens and spec etc of the screen should be mentioned and the possible effects on results discussed in the discussion. This is especially true when screening HIV positive pregnant women - do you know what proportion of the screened women had HIV?

-I was surprised to read under quantitative analysis that you would be comparing rates in pregnant and non-pregnant women as this had not been mentioned before. In the Introduction you mention Zenner et al. who showed higher TB rates in pregnant compared to non-pregnant populations, there are other studies also showing this - please explain to us in the intro why this needs to be done.

- I dont understand why you adjusted the estimate in non-pregnant females to 9 months since this was cross-sectional?

- Under qualitative analysis you say that sample size was decided a priori - what was this based on?

Results:

-What was the HIV status of those with TB?

Discussion:

- In the first paragraph you say the benefits of early detection are clear because of the infant death but that does not make sense to me since this mother was screened but still had a poor outcome?

- I dont understand what you are trying to say with the second paragraph? You say that TB symptoms are hidden in pregnancy (should reference this) but then go on to discuss the rate of cough, which may or may not be due to TB? I also thought that HIV positive women may be even less symptomatic than HIV-negative?

- Last sentence of paragraph 3 - how was this calculated? It should be in the methods and results if you are reporting it.

- Limitations - How would asking about only cough and not other symptoms affect results? I also wonder about the burden of asking about the additional 3 TB symptoms if you are already asking about the cough? How might that affected the cos-effectiveness ratio?

-You say knowledge of TB in pregnancy was low but this was not reflected in your results. In the results all you say is that knowledge of haemoptysis was low which is not really a common symptom. What was the basis for saying this?

- You say the cost per case detected was high compared to other interventions in Benin. What interventions were these?

Reviewer #3: 1- The authors considered data from the Benin NTP with the number of notified cases of TB in 2016 and 2018 where 683 patients were diagnosed (results section) and from which the annual incidence for the study was deduced. In the calculations performed by the authors, were a distinction between pulmonary TB and EPTB data considered in the calculations? If so, mention please it in the materials and methods section, then discuss EPTBs effect in the prevalence observed in the discussions section.

2- Was the obtained TB risk calculated obtained from all the participating pregnant women or is this risk observed only in those with among symptomatic pregnant women (cough lasting two weeks)? This has to be clarified in the manuscript.

3- Limiting the inclusion to a two-week cough alone may be the cause of the lower than estimated outcome (as already cited in the discussions section). Recent studies in another country in sub-Saharan Africa, in an area where the HIV prevalence is lower than that of Benin, but with a higher incidence of TB showed a prevalence of 12.3% of TB in symptomatic pregnant women including coughs lasting more than two weeks but associated with other symptoms such as fever, chest pain, etc. (Ranaivomanana et al., 2021, IJTLD). Please confirm that the other symptoms of TB (fever, chest pain , loss of appetite, etc.) are not considered in the policies of the NTP in Benin as stated in the manuscript?

4- The low prevalence of cough found in the pregnant women of the study seems surprising knowing that the whole immune system of pregnant women is attenuated to protect the foetus. What steps did the authors take to ensure that the cough lasted at least two weeks? Is it by just verbally questionning the participants? How to confirm a two week cough? Please specify it in the manuscript.

5- Since the prices displayed for laboratory tests are those of the GDF, it would be interesting to also give the % of each cost in the table (as in the text). This can be of interest to those who do not have access to the GDF program.

6- Wouldn’ it be interesting to perform a DALY analyzis for this kind of study given the target population? It would be interesting to discuss about this.

6. PLOS authors have the option to publish the peer review history of their article (what does this mean?). If published, this will include your full peer review and any attached files.

Reviewer #1: No

Reviewer #2: **Yes: **Jasantha Odayar

Reviewer #3: No

---

## [Author Response · Author response to Decision Letter 0]

19 Jan 2022

Please see our attached responses.

---

## [Editor Report · Decision Letter 1]

2 Feb 2022

PONE-D-21-23077R1Prevalence, acceptability and cost of routine screening for pulmonary tuberculosis among pregnant women in Cotonou, BeninPLOS ONE

Dear Dr. Campbell,

Thank you for submitting your manuscript to PLOS ONE. After careful consideration, we feel that it has merit but does not fully meet PLOS ONE’s publication criteria as it currently stands. Therefore, we invite you to submit a revised version of the manuscript that addresses the points raised during the review process.

 Thank you for submitting the revised manuscript which is detailed and clear.  1. Both reviewers 1 and 2 noted that screening was conducted at a single time-point during pregnancy. This has been included as a limitation. However, your response to Reviewer 2 point 7 is incomplete:Methods – line 186 on the in the tracked document: The study design is cross-sectional with screening at a single time-point per participant. These data shouldn’t be used to calculate the TB incidence over 9 months for your sample and you are unable to make the comparison with TB incidence over 9 months from the national data (risk difference). You could drop this analysis and amend the relevant lines in the discussion and conclusion. 2. Reporting qualitative results: please indicate whether the quoted responses are from a pregnant woman or midwife.

We look forward to receiving your revised manuscript.

Kind regards,

Emma K. Kalk

Academic Editor

PLOS ONE
---

## [Author Response · Author response to Decision Letter 1]

3 Feb 2022

Please see the attached responses.

---

## [Editor Report · Decision Letter 2]

7 Feb 2022

Prevalence, acceptability, and cost of routine screening for pulmonary tuberculosis among pregnant women in Cotonou, Benin

PONE-D-21-23077R2

Dear Dr. Campbell,

We’re pleased to inform you that your manuscript has been judged scientifically suitable for publication and will be formally accepted for publication once it meets all outstanding technical requirements.

Kind regards,

Emma K. Kalk

Academic Editor

PLOS ONE
---

## [Editor Report · Acceptance letter]

9 Feb 2022

PONE-D-21-23077R2 

Prevalence, acceptability, and cost of routine screening for pulmonary tuberculosis among pregnant women in Cotonou, Benin 

Dear Dr. Campbell:

I'm pleased to inform you that your manuscript has been deemed suitable for publication in PLOS ONE. Congratulations! Your manuscript is now with our production department. 

Kind regards, 

on behalf of

Dr. Emma K. Kalk 

Academic Editor

PLOS ONE